# Astrocytic Regulation of Endocannabinoid-Dependent Synaptic Plasticity in the Dorsolateral Striatum

**DOI:** 10.3390/ijms25010581

**Published:** 2024-01-01

**Authors:** Louise Adermark, Rosita Stomberg, Bo Söderpalm, Mia Ericson

**Affiliations:** 1Department of Pharmacology, Institute of Neuroscience and Physiology, The Sahlgrenska Academy, University of Gothenburg, 40530 Gothenburg, Sweden; 2Department of Psychiatry and Neurochemistry, Institute of Neuroscience and Physiology, The Sahlgrenska Academy, University of Gothenburg, 40530 Gothenburg, Sweden; rosita.stomberg@gu.se (R.S.); bo.soderpalm@gu.se (B.S.); mia.ericson@gu.se (M.E.); 3Beroendekliniken, Sahlgrenska University Hospital, 41345 Gothenburg, Sweden

**Keywords:** astroglia, astrocyte, endocannabinoid system, LTD, synaptic plasticity

## Abstract

Astrocytes are pivotal for synaptic transmission and may also play a role in the induction and expression of synaptic plasticity, including endocannabinoid-mediated long-term depression (eCB-LTD). In the dorsolateral striatum (DLS), eCB signaling plays a major role in balancing excitation and inhibition and promoting habitual learning. The aim of this study was to outline the role of astrocytes in regulating eCB signaling in the DLS. To this end, we employed electrophysiological slice recordings combined with metabolic, chemogenetic and pharmacological approaches in an attempt to selectively suppress astrocyte function. High-frequency stimulation induced eCB-mediated LTD (HFS-LTD) in brain slices from both male and female rats. The metabolic uncoupler fluorocitrate (FC) reduced the probability of transmitter release and depressed synaptic output in a manner that was independent on cannabinoid 1 receptor (CB1R) activation. Fluorocitrate did not affect the LTD induced by the CB1R agonist WIN55,212-2, but enhanced CB1R-dependent HFS-LTD. Reduced neurotransmission and facilitated HFS-LTD were also observed during chemogenetic manipulation using Gi-coupled DREADDs targeting glial fibrillary acidic protein (GFAP)-expressing cells, during the pharmacological inhibition of connexins using carbenoxolone disodium, or during astrocytic glutamate uptake using TFB-TBOA. While pretreatment with the N-methyl-D-aspartate (NMDA) receptor antagonist 2-amino-5-phosphonopentanoic acid (APV) failed to prevent synaptic depression induced by FC, it blocked the facilitation of HFS-LTD. While the lack of tools to disentangle astrocytes from neurons is a major limitation of this study, our data collectively support a role for astrocytes in modulating basal neurotransmission and eCB-mediated synaptic plasticity.

## 1. Introduction

Astrocytes constitute an integral element of the neuronal synapse and provide metabolic and structural support to enable a high signal-to-noise ratio [1,2]. Astrocytes may directly communicate with neurons at the synapse [3,4] and can thus play a role in synchronizing neuronal networks and integrating neurotransmission [5,6,7]. Furthermore, astrocytes are important regulators of synaptic plasticity, including of long-term depression (LTD) mediated by endocannabinoid (eCB) signaling [8]. Endocannabinoids are arachidonoyl metabolites, such as arachidonoyl ethanolamide (anandamide) and 2-arachidonoyl glycerol (2-AG), which derive from the enzymatic breakdown of membrane lipids [9]. In the striatum, eCBs are synthesized on demand in response to elevated intracellular calcium [10,11,12,13] and, following their release, act retrogradely on presynaptic cannabinoid 1 receptors (CB1R) to reduce the probability of transmitter release at excitatory and inhibitory terminals [14,15,16,17,18]. The production and release of striatal eCBs are further regulated by glutamatergic, dopaminergic and acetylcholinergic signaling [19,20,21,22,23,24].

The crosstalk between neurons and astrocytes in eCB-mediated plasticity partially depends on the brain region studied. Astrocytes may regulate eCB signaling through the cellular uptake and hydrolysis of eCBs [25], but they can also produce eCBs as well as express CB1Rs themselves [26,27,28,29,30,31]. The activation of astrocytic CB1Rs may further elicit calcium signaling and promote gliotransmission [32]. In the suprachiasmatic nucleus, eCB signaling induces the release of the gliotransmitter adenosine, thereby reducing the probability for transmitter release from GABAergic terminals [33]. Activation of CB1Rs on cortical or hippocampal astrocytes instead results in glutamate release and elevated levels of glutamate [32,34]. Rat spinal astrocytes co-express CB1R and the 2-AG synthesizing enzyme diacylglycerol lipase-alpha in close proximity and can release 2-AG upon CB1R activation [28], thereby enabling bidirectional signaling that may act to further suppress neurotransmission. In the hippocampus, astrocyte-derived eCBs may also suppress neurotransmitter release without the involvement of gliotransmitters [35].

A major challenge in understanding the role of astrocytes in complex neurophysiological phenomena lies in the limited number of tools available to disentangle astrocytes from neurons. One compound that has been widely used to inhibit astrocytes is the metabolic uncoupler fluorocitrate (FC). Fluorocitrate acts by blocking aconitase, an enzyme utilized in the tricarboxylic acid (TCA) cycle [36]. Local or intraventricular injection of FC has been demonstrated to decrease ATP levels and mitochondrial aconitase activity [37,38] and may further impair astrocytic calcium signaling and the astrocytic clearance of amino acids [39,40,41,42,43]. The specificity of FC may, however, be questioned, and part of its inhibitory property could be attributed to the chelation of extracellular Ca^2+^ and Zn^2+^ produced by the increase in extracellular citrate [36,44]. Another way to selectively target astrocytes is through designer receptors exclusively activated by designer drugs (DREADDs). Through this chemogenetic approach, G-protein coupled receptors can be expressed in astrocytes and then activated by the agonist clozapine N-oxide dihydrochloride (CNO) [45]. It has, however, been questioned if DREADDs expressed in astrocytes accurately mimic endogenous G-protein receptor activity [46], and a high concentration of CNO (1 mM) can induce calcium signaling in both Gq- and Gi/o-expressing astrocytes [47].

The overall aim of this study was to further outline the role of astrocytes in regulating synaptic plasticity in the form of eCB-LTD. Since no manipulation may be considered fully selective, we employed a battery of tools previously engaged to suppress astrocyte function. Astrocytic activity was targeted by metabolic, chemogenetic and pharmacological approaches, and synaptic activity and stimulation-induced plasticity were assessed by ex vivo electrophysiology. Recordings were conducted in the dorsolateral striatum (DLS), an important brain region for the consolidation of motor skill learning and the formation of habits [48,49].

## 2. Results

### 2.1. Impaired Astrocyte Function Enhances Endocannabinoid-Mediated Synaptic Plasticity at Excitatory Synapses in the Striatum 

High-frequency stimulation induced CB1R-dependent LTD (HFS-LTD) in the DLS of both male and female rats (male vs. female: F_(1,13)_ = 0.04751, *p* = 0.8308; combined group vs. AM251: F_(1,20)_ = 12.89, *p* = 0.0018) (Figure 1B,C). HFS-LTD coincided with an increase in the PPR (t_(13)_ = 4.622, *p* < 0.001) (Figure 1D), indicating that HFS decreases synaptic output by reducing the probability of transmitter release.

In an attempt to outline the role of astrocytes in regulating HFS-LTD, slices were pretreated with the glia-specific TCA-cycle inhibitor FC prior to HFS. Fluorocitrate both depressed evoked populations spikes in field-potential recordings (t_(22)_ = 6.558, *p* < 0.001) and suppressed the frequency of recorded sEPSCs (t_(9)_ = 5.533, *p* < 0.001) (Figure 1F), which is in agreement with previous findings [39]. Synaptic depression induced by FC was not inhibited by the CB1R antagonist AM251 (F_(1,45)_ = 0.03380, *p* = 0.8550) (Figure 1G). However, HFS-LTD was facilitated in slices continuously perfused with FC (F_(1,13)_ = 21.70, *p* < 0.001) (Figure 1H). AM251 blocked LTD in FC-treated slices, suggesting that the potentiation of HFS-LTD is associated with eCB signaling (Figure 1I). Importantly, synaptic depression induced by the CB1R agonist WIN55,212-2 (2 μM) was not facilitated by FC (F_(1,35)_ = 0.01563, *p* = 0.9012) (Figure 1J), indicating that astrocytic regulation of eCB signaling is upstream from presynaptic CB1R activation.

### 2.2. Activation of Gi-Coupled DREADDs Targeting Astrocytes Facilitates HFS-LTD 

In the next set of experiments, Gi- and Gq-coupled DREADDs targeting GFAP-expressing cells were used in another attempt to selectively manipulate astrocytic activity. Three weeks after viral transfection into the DLS, field-potential recordings were conducted in close vicinity to the end of the faint scar (Figure 2B). Bath-perfused CNO (10 μM) slightly depressed the PS amplitude in all treatment groups (Figure 2C) and significantly enhanced HFS-LTD selectively in slices expressing Gi-coupled DREADDs targeting GFAP (F_(2,32)_ = 5.819, *p* = 0.0070; scrambled control vs. Gi-DREADDs: *p* = 0.0130) (Figure 2C,D). Since 10 μM CNO also depressed the PS amplitude in the sham-treated control, we applied a lower concentration to reduce putative unspecific effects by CNO/DMSO. A lower concentration of CNO (2 μM) selectively depressed the PS amplitude in brain slices from rats expressing Gi-DREADDs (one-way ANOVA: F_(3,74)_ = 5.315, *p* = 0.0023; aCSF vs. Gi-DREADDs: *p* = 0.0070) (Figure 2E) and enhanced HFS-LTD (F_(2,31)_ = 5.610, *p* = 0.0083; Sham vs. Gi-DREADDs: *p* = 0.028) (Figure 2G). HFS-LTD was not affected in brain slices from rats expressing Gq-coupled DREADDs compared with sham-treated controls (Figure 2G). The PPR was significantly enhanced by HFS in all treatment groups (Sham: t_(11)_ = 3.264, *p* = 0.0075; Gq-DREADDs: t_(12)_ = 3.260, *p* = 0.0075; Gi-DREADDs: t_(15)_ = 3.279, *p* = 0.0051) (Figure 2H). 

### 2.3. Impaired Gap-Junction Coupling Enhances HFS-LTD

Striatal astrocytes are extensively interconnected via gap junctions, and the astroglial syncytium enables the transport of different substrates of importance for neurotransmission. In the next set of experiments, slices were treated with the gap-junction blocker carbenoxolone disodium (100 µM), which rapidly suppresses cell-to-cell coupling between striatal astrocytes [50]. Carbenoxolone disodium significantly depressed the PS amplitude by itself (F_(1,31)_ = 16.9, *p* < 0.001) (Figure 3A) and increased the PPR (t_(13)_ = 5.587, *p* < 0.001) (Figure 3B), suggesting that synaptic depression is concomitant with a reduced probability of transmitter release. 

Continuous perfusion of carbenoxolone disodium facilitated HFS-LTD (F_(1,29)_ = 25.88, *p* < 0.001) (Figure 3C). This finding was further confirmed by whole-cell recordings, demonstrating an enhanced depression of excitatory postsynaptic currents in slices perfused with a gap-junction coupling inhibitor during application of the HFS protocol (F_(1,9)_ = 32.6, *p* < 0.001) (Figure 3D). Synaptic depression induced by the CB1R agonist WIN55,212-2 (2 µM) was not facilitated in slices pretreated with carbenoxolone disodium for forty minutes prior to WIN55,212-2 (t_(18)_ = 0.5571, *p* = 0.5571; Figure 3E).

### 2.4. Increased Glutamatergic Neurotransmission May Underly the Facilitation of HFS-LTD

Glutamatergic neurotransmission is pivotal for postsynaptic eCB production and release [19,22], and astrocytes are key regulators of extracellular glutamate. To assess if elevated glutamate levels could underly the facilitation of HFS-LTD, slices were perfused with the glial-specific glutamate-transporter inhibitor TFB-TBOA (200 nM). TFB-TBOA depressed the evoked PS amplitude in a sex-independent manner (F_(1,31)_ = 0.2982, *p* = 0.5889) (Figure 4A) and facilitated HFS-LTD (F_(1,29)_ = 9.755, *p* = 0.0040) (Figure 4B), suggesting that increased glutamate signaling caused by impaired astrocyte function could act to enhance striatal eCB signaling. Perfusion for 30 min with TFB-TBOA prior to WIN55,212-2 and continuously throughout the experiment did not affect the synaptic depression induced by CB1R activation (F_(1,24)_ = 0.03370, *p* = 0.8559) (Figure 4C).

Increased levels of extrasynaptic glutamate as a result of impaired glutamate buffering may also activate metabotropic glutamate receptor 2/3 (mGluR2/3), which, upon activation, could act to depress the probability of transmitter release [20]. However, pretreatment with the mGluR2/3 antagonist LY 341495 did not prevent FC-mediated synaptic depression (aCSF vs. LY 341495 +FC: F_(1,47)_ = 16.03, *p* < 0.001) (Figure 4D). Importantly, in addition to glutamate clearance, astrocytes may release kynurenic acid, which may act to tonically inhibit NMDA receptors [52]. NMDA receptors are highly permeable to calcium and may thus promote eCB production upon activation [10,22]. Fluorocitrate has previously been shown to robustly reduce the levels of kynurenic acid in the prefrontal cortex [51]; thus, in the last set of experiments, slices were pretreated with the NMDA receptor antagonist APV (50 µM) prior to FC administration and HFS. Pretreatment with APV was not sufficient to prevent the synaptic depression induced by FC (aCSF vs. APV + FC: F_(1,62)_ = 19.48, *p* < 0.001; FC vs. APV + FC: F_(1,14)_ = 0.6560, *p* = 0.4315; F_(1,62)_ = 19.48, *p* < 0.001; PPR: t_(14)_ = 3.985, *p* = 0.0014) (Figure 4D,E). However, inhibition of NMDA receptors blocked the potentiating effect by FC on HFS-LTD (F_(1,26)_ = 0.1000, *p* = 0.7543) (Figure 4G,H), supporting the idea that astrocytes may influence striatal eCB signaling through the regulation of glutamatergic neurotransmission. 

## 3. Discussion

The overall aim in this study was to further outline the role of astrocytes in regulating the striatal LTD mediated by eCBs. While the properties of the astrocytes make them hard to disentangle from neurons, and none of the applied approaches can be stated to inhibit astrocyte function specifically, all the tools employed generated similar results, independent of whether astrocyte function was impaired using metabolic, chemogenetic or pharmacological approaches. Overall, our data suggest that astrocytes are important regulators of synaptic activity in the striatum and may play an active role in modulating synaptic plasticity in the form of eCB-mediated LTD. Importantly, inhibition of astrocytes only facilitated HFS-LTD and did not promote LTD induced by the CB1R agonist. This indicates that eCB signaling is potentiated at a step that is upstream from CB1R activation and is possibly linked to increased levels of circulating eCBs.

In the first set of experiments, astrocyte function was impaired when the metabolic uncoupler FC was used [36]. Fluorocitrate substitutes citrate in the TCA cycle and inhibits glutamine formation from the glial-specific substrate ^[14C]^acetate while having no effect on the metabolism of ^[14C]^glucose, which is utilized by neurons [53,54,55]. Fluorocitrate has repeatedly been used to impair astrocyte function in both cell studies and behavioral studies [40,41,42,43,56,57], and we have previously demonstrated that striatal administration of FC robustly elevates extracellular glutamate and progressively depresses glutamine levels, suggesting that FC impairs the glutamate–glutamine shuttle [39]. Impaired glutamate clearance may be related to FC-mediated inhibition of the formation of energy sources such as ATP [37], but reduced calcium signaling and changes in intracellular ion gradients may also contribute to a reduced glutamate uptake. There are also studies showing that FC suppresses glutamine synthase gene expression [58], which may further affect glutamine levels during continuous administration. Furthermore, the effect by FC is dose-dependent [59], and the concentration used here may only have minor effects on ATP levels [60].

Even though extracellular glutamate is elevated by FC [39], the evoked field potentials and the frequency of spontaneous excitatory currents were depressed. Importantly, this synaptic depression was not CB1R-dependent, demonstrating that the decrease in PS amplitude elicited by bath-perfusion of FC is not mediated through the same signaling pathways as stimulation-induced LTD. The decrease in PS amplitude was further insensitive to antagonists targeting mGluR2/3 or NMDA receptors. While FC-mediated synaptic depression could be connected to unspecific effects by FC, such as the chelation of extracellular Ca^2+^ [36], it has previously been demonstrated to be prevented by pretreatment with a dopamine D2-receptor antagonist. In fact, local perfusion of FC in the DLS not only increases glutamate levels but also raises dopamine, thereby producing a complex impact on striatal neurotransmission [39]. Importantly, while hippocampal astrocytes have been acknowledged to regulate synaptic plasticity through the release of serine and glycine [6,61], these amino acids do not appear to be affected by FC in the DLS [39]. Since striatal astrocytes display more passive properties in comparison with hippocampal astrocytes [50], it is possible that astrocytes exhibit brain-region-specific functions. It is also possible that D-serine is primarily released from striatal neurons, which are not affected by FC [62].

Even though evoked potentials were robustly depressed by FC, HFS-LTD was further enhanced, indicating that inhibition of astrocyte function using FC promotes eCB signaling and the induction of LTD. Importantly, the CB1R antagonist blocked HFS-induced LTD in both aCSF-treated controls and FC-exposed slices, suggesting that FC does not induce LTD through an alternative pathway that does not involve eCB signaling [63]. Since synaptic depression induced by WIN55,212-2 was not potentiated in slices pretreated with FC, the regulatory properties of astrocytes are most likely connected to enhanced eCB synthesis and/or release or the reduced degradation of eCBs [14,64].

In the next set of experiments, astrocytes were targeted using chemogenetic approaches. Striatal cells were transfected with Gi- or Gq-coupled DREADDs targeting GFAP and perfused with the agonist CNO. Immunohistological staining demonstrated that DREADD expression was associated with GFAP, and the main area of expression was relatively restricted to the injection site. While a higher concentration of CNO (10 µM) slightly depressed evoked PS amplitudes in all treatment groups, a lower concentration (2 µM) selectively inhibited PS amplitudes in brain slices expressing Gi-coupled DREADDs. While DMSO could have affected neurotransmission, the proportion was only 0.05%. It is thus possible that higher doses of CNO may produce unspecific effects when applied in combination with electrical stimulation. Perfusion of 2 µM CNO selectively depressed the PS amplitude in brain slices from rats transfected with Gi-coupled DREADDs, but the relative depression was not as pronounced compared with FC and carbenoxolone disodium. Nonetheless, Gi-coupled DREADDs were associated with a robust enhancement of HFS-LTD, supporting the idea that the mechanisms triggering synaptic depression during drug perfusion (baseline conditions) may differ from the signaling pathways recruited during the facilitation of HFS-LTD [40]. It should be noted that the ability for DREADDs expressed in astrocytes to accurately mimic endogenous G-protein receptor activity has been discussed [46]. In addition, the capacity to activate or deactivate astrocytes via G-protein-coupled receptors have also been questioned [45,46]. However, while the activation of both astrocytic Gi- and Gq-coupled receptors may increase intracellular [Ca^2+^]_i_ in astrocytes, the downstream signaling pathways, including phospholipase C and protein kinase A, still appear to be sufficient to produce specific and functional outputs [65,66,67,68].

Astrocytic gap-junction channels are primarily built up by connexin hemichannels, and these channels are crucial for the synchronization of astrocytic signaling, calcium wave propagation and the spatial buffering of potassium [69,70]. Connexin hemichannels have also been suggested to be involved in the release of gliotransmitters and biomolecules from astrocytes [71,72], and they are required for the diffusion of metabolites such as glucose and lactate [73,74]. Bath perfusion of carbenoxolone disodium robustly depressed synaptic activity and increased the PPR, indicating that inhibition of gap-junction coupling reduces the probability of transmitter release at excitatory synapses. Carbenoxolone disodium further facilitated HFS-LTD at excitatory synapses, as demonstrated by both field-potential and whole-cell recordings, further supporting a role for astrocytes in modulating neurotransmission [75]. Carbenoxolone disodium should, however, not be considered selective for astrocytic gap junctions, and electrical coupling also exists, to some extent, between striatal inhibitory fast-spiking interneurons [76]. In addition, regular carbenoxolone dissolved in DMSO has been reported to inhibit inositol 1,4,5-trisphosphate-mediated endothelial-cell calcium signaling, which depolarizes mitochondria and blocks calcium channels [77,78]. While a water-soluble gap-junction blocker was used in the experiments presented here, carbenoxolone disodium is a glucocorticoid that also inhibits 11 β-hydroxysteroid dehydrogenase. Additional experiments using other gap-junction blockers such as connexin-mimetic peptides or antibodies would thus be warranted to confirm these findings.

Inhibition of astrocytic glutamate transporters also decreased the PS amplitude and facilitated HFS-LTD. This was somewhat surprising since we have previously shown that mice with a genetic deletion of the astrocyte-specific glutamate transporter GLAST exhibit impaired eCB production and release at excitatory synapses in the striatum, with no effect on WIN55,212-2-induced synaptic depression [79]. This finding may be connected to neuroadaptations elicited by the genetic deletion. For instance, baseline glutamate levels were not significantly elevated in GLAST KO mice [79], but they are rapidly enhanced during acute impairment of astrocyte function [39]. It is thus possible that GLAST KO mice exhibit enhanced glutamate uptake through GLT-1 or display other neuroadaptations that counterbalance the eCB-LTD-promoting effect.

Several signaling pathways are involved in eCB signaling and LTD at striatal synapses. Calcium needs to be raised in the MSNs and a certain amount of synaptic activity is required for eCBs to be released and LTD to be induced [10,11]. In particular, glutamate receptors appear to play a key role in regulating eCB signaling. Activation of mGluR group 1 receptors promotes eCB-mediated LTD [21,80], and stimulation-induced release of 2-AG is robustly depressed by antagonists targeting AMPA or NMDA receptors [22]. Importantly, astrocytes are not only important for clearing glutamate from the extracellular space, but they also release bioactive molecules that may affect glutamate signaling [51,81]. In particular, the release of the glia-derived molecule kynurenic acid may be important, considering its inhibitory effect on ionotropic glutamate receptors [52,82]. During the inhibition of astrocytes, extracellular glutamate will increase, leading to enhanced activation of excitatory receptors. In addition, the glial-associated release of kynurenic acid may decrease [51], further resulting in an increased influx of calcium though ligand-gated glutamate receptors. In addition, during incubation with FC, increased levels of citrate will most likely result in the increased chelation of Zn^2+^, thereby further increasing NMDA receptor activation [36]. While striatal LTD can occur independently of NMDA receptor signaling [83], the high permeability to calcium may substantially enhance the increase in postsynaptic calcium levels and thus facilitate eCB production and release [22]. Further supporting this theory, we found that pretreatment with the NMDA receptor antagonist APV prevented the FC-mediated potentiation of HFS-LTD. Thus, while astrocytes may affect multiple signaling pathways that could promote eCB signaling and the formation of LTD, activation of NMDA receptors appears to play a key role. Considering that the intracellular loading of BAPTA in the postsynaptic neuron prevents eCB-LTD [11], we postulate that the release of eCBs primarily derives from neurons, but it is also possible that eCBs are released from astrocytes in response to increased glutamate signaling [35]. It is also possible that the CB1R activation of astrocytes results in gliotransmission, resulting in the activation of NMDA receptors [84]. Interestingly, an NMDA receptor antagonist further blocks changes in the striatal glucose metabolism elicited by FC, indicating that NMDA receptors also play a role in regulating the metabolic effects by FC [85]. Further research is required to fully outline these associations.

It should be noted that astrocytes have also been suggested to promote HFS-LTD through other signaling pathways. Studies performed in mice have demonstrated that HFS leads to glutamate-induced calcium signaling in astrocytes and release of the gliotransmitter adenosine, thus resulting in synaptic depression through adenosine A1 receptor activation [63]. Since the data presented here required CB1R activation, it is possible that the signaling pathways recruited during HFS-LTD depend on the model system used [63,86]. Another possible neurotransmitter involved in the facilitation of HFS-LTD is dopamine. Activation of dopamine D2 receptors is a prerequisite for eCB-LTD [21], and FC has previously been reported to increase dopamine levels in the DLS [39]. While an NMDA receptor antagonist blocked the FC-mediated facilitation of LTD, it is possible that activation of NMDA receptors promotes the increase in dopamine levels or that a concomitant activation is required, but further experiments are needed to outline this interaction. Lastly, astrocytes may regulate both glutamatergic and dopaminergic neurotransmission through the uptake and release of GABA [87,88,89,90,91,92]. While FC-mediated synaptic depression in the striatum is independent of GABA_A_-receptor activation [39], GABAergic neurotransmission may still play a role in the facilitation of HFS-LTD. The impact displayed by changes in the neurometabolic coupling between astrocytes and neurons should also be further explored.

In this study, we applied different methods that have been widely used to inhibit astrocyte function. The metabolic uncoupler FC particularly targets astrocytes [36,54], and a robust impairment of astrocytic function is also supported by in vivo microdialysis data and brain-slice studies [39,43,91]. However, some of the results may be connected to unspecific effects on astrocytes and neurons. Likewise, even though carbenoxolone disodium prevents astrocytic gap-junction permeability [50,93], it may have unspecific side effects and will not act selectively on astrocytes. The low dose of TFB-TBOA should specifically inhibit GLT-1 and GLAST, but GLT-1 has also been detected in neurons in culture [94]. Lastly, Gi-coupled DREADDs targeting GFAP are only expressed in a restricted area and the ability to activate or deactivate astrocytes via G-protein-coupled receptors have been questioned [45]. Still, each approach employed produced similar effects on striatal neurotransmission and plasticity. In conclusion, while the tools available to disentangle astrocytes from neurons are limited, and the methods used in this study cannot be considered fully specific, the data retrieved collectively support an active role for astrocytes in modulating neurotransmission and synaptic plasticity in the form of eCB-LTD. The effects appear to be especially linked to changes in glutamatergic neurotransmission, which, in combination with alterations in striatal eCB signaling, could have a major impact on striatal-based learning and behavior (Augustin et al., 2023; Hilario et al., 2007; Rueda-Orozco et al., 2008) [95,96,97].

## 4. Materials and Methods

### 4.1. Animals

Male Wistar Han rats (Envigo, Horst, The Netherlands), weighing 160–180 g at arrival, were group-housed with a 12 h dark/light cycle and ad libitum access to water and food. Rats were allowed to adapt to the novel environmental conditions (room temperature of 20 °C, relative humidity 65%, and a regular light–dark cycle with lights on at 07:00 a.m. and off at 07:00 p.m.) for at least one week prior to any procedures. All surgeries and experiments were performed during the light phase of the cycle. The experiments were approved by the Ethics Committee for Animal Experiments, Gothenburg, Sweden.

### 4.2. Drugs

The metabolic uncoupler DL-fluorocitric acid barium salt (FC) was dissolved in artificial cerebrospinal fluid (aCSF) containing (in mM) 124 NaCl, 4.5 KCl, 2 CaCl_2_, 1 MgCl_2_, 26 NaHCO_3_, 1.2 NaH_2_PO_4_ and 10 D-glucose at a final concentration of 5 µM. The DREADDs agonist CNO was dissolved in dimethyl sulfoxide (DMSO) to 20 mM and further diluted to 10 or 2 µM in aCSF. For the pharmacological regulation of glutamatergic neurotransmission, the NMDA receptor antagonist DL-2-amino-5-phosphonopentanoic acid (APV) was dissolved in H_2_O to 25 mM and used at 50 µM, while the glutamate-transporter inhibitor TFB-TBOA was dissolved in DMSO to 10 mM and further diluted in aCSF to 200 nM. The CB1R antagonist AM251 and the agonist WIN55,212-2 were dissolved in DMSO to 20 mM and diluted in aCSF to 2 µM. The gap-junction inhibitor carbenoxolone disodium was dissolved in H_2_O (100 mM) and diluted in aCSF to a final concentration of 100 µM. Drugs were purchased from Sigma Aldrich (Stockholm, Sweden) or Tocris (Bristol, UK). 

### 4.3. Surgery for Viral Injections 

Animals were anesthetized with 4% isoflurane (Forene Baxter, Kista, Sweden) and placed in a stereotactic frame (David Kopf Instruments, Tujunga, CA, USA) on a heating pad. The skull was exposed, and one hole was drilled unilaterally above the DLS target area. A 10 μL Hamilton syringe attached to a 31-gauge microinjection canula (AMI-5T, AgnTho’s AB, Lidingö, Sweden) was used for the administration of the virus. The canula was gently lowered into the DLS (A/P: +0.5; M/L: −3.5 relative to bregma; V/D: −5.0 relative to skull) and the solutions were infused using a microperfusion pump (Univentor-864 Syringe Pump; AgnTho’s AB, Lidingö, Sweden). A total volume of 0.8 μL was infused at a flow rate of 0.05 μL/min. To allow for diffusion, the cannula was withdrawn 5 min after completion of the infusion. Injected viruses included pssAAV-2-hGFAP-HA_hM_3_D(Gq)-IRES-mCitrine-WPRE-hGHp(A) (GFAP-Gq-DREADD), pssAAV-5/2-hGFAP-hM_4_D(Gi)_mCherry-WPRE-hGHp(A) (GFAP-Gi-DREADD) and pssAAV-2-hGFAP-EGFP-WPRE-hGHp(A) (sham) (Viral Vector Facility (VVF), Neuroscience Center Zurich (ZNZ), Zurich, Switzerland). The wound was closed using staples. Rats were given postoperative analgesia (Norocarp, 5 mg/kg, s.c.) and returned to their home cage for three weeks prior to electrophysiology experiments or immunohistochemical verification. 

### 4.4. Ex Vivo Electrophysiology

#### 4.4.1. Brain Slice Preparation

Acutely isolated brain slices were prepared as previously described [98]. In brief, rats were anesthetized with isoflurane and decapitated. Brains were quickly removed and submerged in ice-cold modified aCSF containing (in mM) 220 sucrose, 2 KCl, 1.3 NaH_2_PO_4,_ 6 MgCl_2_, 0.2 CaCl_2_, 26 NaHCO_3_ and 10 D-glucose and kept continuously oxygenated (95% O_2_, 5% CO_2_). Coronal brain slices (300 µm) were sectioned and transferred to regular aCSF and kept continuously oxygenated with 95% O_2_ and 5% CO_2_. Slices were allowed to equilibrate at 30 °C for 30 min and then maintained at room temperature for the remainder of the day. No experiment was executed more than seven hours after the brain slice preparation.

#### 4.4.2. Field-Potential Recordings

Evoked population spike (PS) amplitudes in striatal ex vivo field-potential recordings primarily reflect the activation of AMPA receptors, and recordings were performed as previously described in detail [99]. In brief, one hemisphere of the slice constantly perfused with oxygenized aCSF (30 °C, 2 mL/min), and local PSs were evoked with a 20 s interpulse interval using a monopolar stimulating electrode (World Precision Instruments, Sarasota, FL, USA, type TM33B). Evoked responses were registered with a recording electrode positioned in the DLS, and stimulation electrodes were positioned dorsally, 0.2–0.3 mm from the recording electrode. A stable baseline was recorded for at least ten minutes prior to pharmacological manipulation or the induction of eCB signaling. To induce eCB-LTD, four trains of 100 pulses were delivered at 100 Hz with a 10 s inter-train interval [19]. In experiments where multiple manipulations were performed, the initial drug was bath-perfused for at least thirty minutes and continuously throughout the experiment before other agents/stimulation protocols were applied. For experiments involving FC, slices were pretreated for at least 60 min before the initiation of other protocols. To assess changes in the release probability, responses were evoked using a paired-pulse stimulation protocol (0.1 Hz, 50 ms interpulse interval), and the paired-pulse ratio (PPR) was calculated by dividing the second pulse (PS2) with the first pulse (PS1). Field-potential responses smaller than 0.25 mV were not included in the data analysis.

#### 4.4.3. Whole-Cell Recordings

Whole-cell patch-clamp recordings assessing spontaneous excitatory postsynaptic currents (sEPSCs) in the voltage-clamp mode were performed as previously described [98]. In brief, brain slices under a constant flow of preheated aCSF (33 °C, 2 mL/min) and MSNs were clamped at −65 mV using a MultiClamp 700B amplifier (Molecular Devices, Axon CNS, San Jose, CA, USA). The internal solution contained (in mM) 135 K-Glu, 20 KCl, 2 MgCl_2_, 0.1 EGTA, 10 Hepes, 2 Mg-ATP and 0.3 Na-GTP, with the pH adjusted to 7.3 with KOH, and the osmolarity was set to 295 mOsm using sucrose. 

In one experiment, evoked excitatory postsynaptic currents (EPSCs) were measured in the conventional ruptured-patch whole-cell mode in MSNs voltage-clamped at −70 mV, as previously described [10]. In this experiment, the internal solution consisted of (in mm) 120 CsMeSO_3_, 5 NaCl, 10 TEA-Cl, 10 HEPES, 5 QX-314, 1.1 EGTA, 4 Mg-ATP and 0.3 Na-GTP, with the pH adjusted to 7.2 with CsOH. To evoke baseline synaptic currents, a paired stimuli with a 50 ms interpulse interval was delivered every 20 s via an electrode placed in the overlying white matter. Stimulation was delivered by a Master-8 stimulator and an optical stimulus isolation unit (A.M.P.I., Jerusalem, Israel). Stimulus parameters were adjusted to elicit baseline EPSC amplitudes between 200 and 400 pA. To induce eCB signaling, a high-frequency stimulation (HFS) protocol, consisting of four trains of 100 pulses delivered at 100 Hz with a 10 s inter-train interval, was paired with depolarization of the postsynaptic cell to 0 mV. Only whole-cell recordings with a stable series resistance that varied <20% and that did not exceed 25 MΩ were included in the analysis. 

### 4.5. Immunohistochemsitry

Immunohistochemistry was performed as previously described in detail [39]. In brief, brains were fixated in 4% paraformaldehyde (PFA), incubated in sucrose solution and snap-frozen using isopentane on dry ice. Coronal brain slices (40 µm) were cut using a Leica CM1950 cryostat and placed in a cryoprotective solution (30% glycerol, 30% ethylene glycol, 40% 1× Tris Buffered Saline (TBS; 0.15 M NaCl, 0.05M Tris-HCL; pH 7.6)). Brain slices were incubated with monoclonal mouse anti-Glial Fibrillary Acidic Protein 1:1000 (Invitrogen, Cat no: MA512023, Thermo Fischer Scientific, Waltham, MA, USA) and monoclonal rabbit anti-NeuN 1:400 (Cat no: MABN140, Merck Millipore, Darmstadt, Germany) in TBS with 5% normal donkey serum and 0.1% triton X-100 overnight at 4 °C. Slices were washed in TBS and incubated with secondary antibodies, donkey anti-mouse Alexa 555 and donkey anti-rabbit Alexa 488 (1:1000, Invitrogen, Waltham, MA, USA), diluted in TBS with 5% normal donkey serum and 0.1% Triton X-100 for 1 h, followed by a TBS-wash. A reporter molecule, mCherry, was fused to the construct coding for the DREADDs. Thus, no staining or additional enhancement was required to visualize the expression of DREADDs. Brain slices were mounted onto microscope slides using Fluoroshield (Sigma-Aldrich, St. Louis, MO, USA), and images were obtained using a Zeiss LSM 700 inverted confocal microscope (Zeiss, Jena, Germany).

### 4.6. Statistics

Data were analyzed using Clampfit 10.2 (Molecular Devices, Axon CNS, San Jose, CA, USA), Microsoft Excel 16.79.2 (Redmond, WA, USA) and GraphPad Prism 10 (GraphPad Software, San Diego, CA, USA). Gaussian distribution was tested with the D’Agostino–Pearson omnibus normality test. For the statistical analysis of data containing consecutive time points, 2-way analysis of variance (ANOVA) was applied. For data presented in bar graphs, one-way ANOVA or *t*-tests were applied when appropriate. All parameters are given as mean ± SEM, and the level of significance was set to *p* < 0.05.

## Figures and Tables

**Figure 1 ijms-25-00581-f001:**
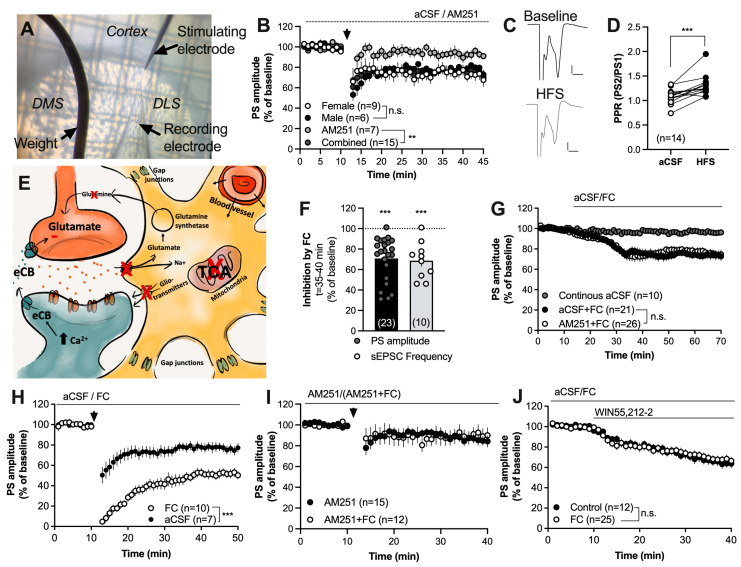
Inhibition of the TCA cycle in astrocytes facilitates striatal HFS-LTD. (**A**) Micrograph demonstrating the position of recording and stimulating electrodes in the DLS. (**B**) HFS induced LTD in brain slices from both female and male rats in a CB1R-dependent manner. (**C**) Example traces demonstrating evoked field potentials at baseline (black) and after HFS (gray). Calibration: 0.2 mV, 2 ms. (**D**) The PPR was significantly enhanced following HFS, indicating that synaptic depression is associated with a decrease in the probability of transmitter release. (**E**) Schematic and simplified drawing of the tripartite synapse. Striatal astrocytes are extensively interconnected through gap junction channels and form a syncytium that may even include cortical astrocytes. Astrocytes may regulate neurotransmission through the clearance of glutamate by releasing gliotransmitters and by influencing neurovascular coupling. Inhibition of the TCA-cycle in astrocytes using the metabolic uncoupler FC has been demonstrated to decrease the clearance of glutamate resulting in reduced release of the substrate glutamine. Incubation with FC further leads to reduced release of gliotransmitters and bioactive molecules, and impaired astrocytic calcium signaling [39,40,50,51]. (**F**) Perfusion of FC (5 μM) for 30 min depressed the evoked PS amplitudes and reduced the frequency of glutamatergic inputs onto MSNs. (**G**) Synaptic depression induced by FC was not CB1R-dependent. (**H**) Facilitation of HFS-LTD was observed in slices pretreated with FC for at least 60 min and continuously throughout the experiment. (**I**) The CB1R antagonist prevented HFS-LTD in both aCSF- and FC-treated slices, suggesting that facilitation of LTD is associated with eCB signaling. (**J**) Synaptic depression induced by the CB1R agonist was not enhanced during the metabolic inhibition of astrocytes. Data are mean values ± sem. Arrows mark time-points for HFS. n = number of recordings, taken from at least four animals. n.s. = not significant. ** *p* < 0.01; *** *p* < 0.001.

**Figure 2 ijms-25-00581-f002:**
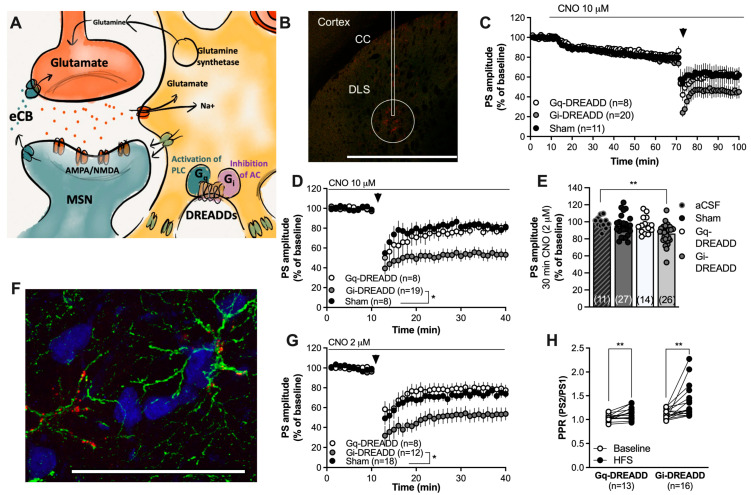
Chemogenetic activation of Gi-coupled DREADDs in astrocytes facilitates eCB-LTD. (**A**) Schematic drawing depicting the use of Gq- or Gi-coupled DREADDs to activate pathways that putatively could stimulate or depress astrocytic function. (**B**) Viral transfection of the DREADD virus was visualized with mCherry (red). The rectangle marks the approximate position of the cannula track; note that there was no damage to the brain tissue following viral injection. While mCherry could be visualized in some cells surrounding the cannula track and even in cells located in the cortex, the circle demonstrates the main area of transfection. Calibration: 1 mm. CC: corpus callosum. (**C**) Perfusion with CNO (10 μM) slightly depressed the PS amplitude in all treatment groups. (**D**) Re-baselining-evoked PS amplitudes revealed an enhanced HFS-LTD in CNO-perfused slices expressing Gi-DREADDs targeting GFAP. (**E**) Bar graph demonstrating the PS amplitudes following bath perfusion with CNO (2 μM) for 30 min. A significant depression of evoked potentials was selectively observed in brain slices expressing Gi-DREADDs. (**F**) Immunohistochemical staining showing GFAP-expressing astrocytes in green, neurons visualized by NeuN in blue, and the DREADD conjugated with mCherry in red. mCherry was associated with cells expressing GFAP. Calibration: 50 µm. (**G**) Pretreatment with a lower dose of CNO still facilitated HFS-LTD in slices transfected with Gi-DREADDs targeting GFAP compared to the control. (**H**) The PPR was enhanced following HFS in both Gq-DREADD- and Gi-DREADD-expressing slices, indicative of the reduced probability for transmitter release. Data are mean values ± sem. Arrows mark time-points for HFS. n = number of recordings, taken from at least four animals. * *p <* 0.05; ** *p* < 0.01.

**Figure 3 ijms-25-00581-f003:**
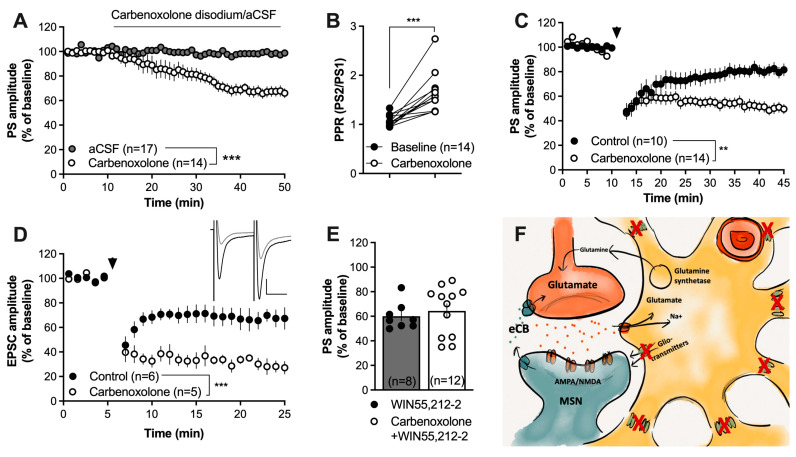
Blockade of connexins facilitates HFS-LTD. (**A**,**B**) The gap-junction uncoupler carbenoxolone disodium (100 µM) significantly depressed the evoked PS amplitude and increased the PPR. (**C**) Pretreatment with carbenoxolone enhanced HFS-LTD. (**D**) Carbenoxolone-induced facilitation of HFS-LTD at excitatory synapses was confirmed in whole-cell recordings. Note the evoked EPSCs during carbenoxolone perfusion (black) and after HFS (gray) in the upper right panel. Calibration: 100 pA, 25 ms. (**E**) Carbenoxolone disodium did not facilitate the synaptic depression induced by the CB1R agonist WIN55,212-2 (2 µM). (**F**) Schematic drawing demonstrating how carbenoxolone treatment would prevent gap-junction coupling and putatively impair the release of gliotransmitters via hemichannels built up by connexins. Data are presented as mean values ± SEM. Arrows mark time-points for HFS. n = number of recordings, taken from at least three animals. ** *p* < 0.01, *** *p* < 0.001.

**Figure 4 ijms-25-00581-f004:**
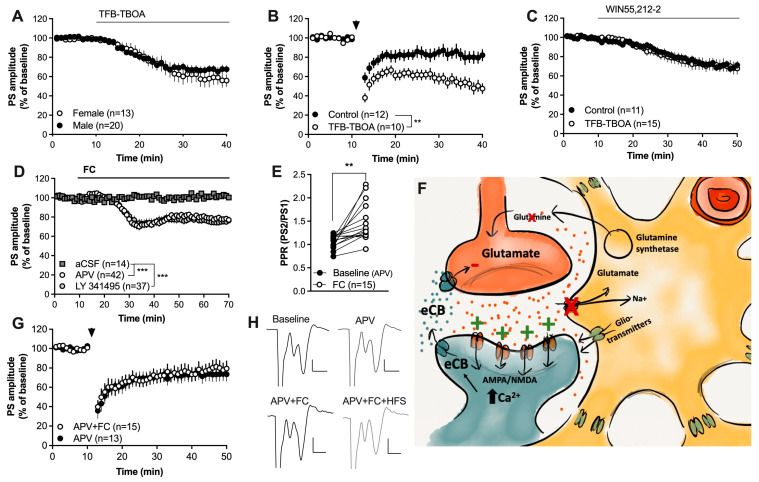
Enhanced glutamatergic signaling may underly the facilitation of HFS-LTD. (**A**) Inhibition of astrocytic glutamate uptake produced synaptic depression in brain slices from both male and female rats. (**B**) Slices pretreated with TFB-TBOA were more susceptible to HFS-LTD. (**C**) TFB-TBOA did not influence the synaptic depression induced by the CB1R agonist. (**D**) Neither the mGluR2/3 antagonist LY 341495 nor the NMDA receptor antagonist APV prevented the synaptic depression induced by FC. (**E**) The PPR was significantly enhanced by FC in slices pretreated by FC. (**F**) Schematic drawing depicting a putative mechanism for how astrocytes may regulate eCB signaling. Inhibition of astrocyte function not only results in impaired glutamate clearance and reduced release of the substrate glutamine, but also the reduced release of gliotransmitters and bioactive molecules, including kynurenic acid, which normally would suppress NMDA receptor activation. Increased levels of glutamate in combination with reduced inhibition of NMDA receptors could lead to increased postsynaptic activation of glutamatergic receptors, resulting in increased postsynaptic calcium, which would promote eCB production and release. (**G**) APV blocked the ability of FC to promote HFS-LTD. (**H**) Example traces demonstrating evoked potentials at baseline and following APV (upper traces) and in other slices during incubation in APV + FC followed by HFS (lower traces). Calibration: 0.2 mV, 2 ms. Data are presented as mean values ± SEM. n = number of recordings, taken from at least four animals. Arrows mark time-points for HFS. ** *p* < 0.01; *** *p* < 0.001.

## Data Availability

The data presented in this study are available on request from the corresponding author (L.A).

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
