# Peer review of "Astrocytic Regulation of Endocannabinoid-Dependent Synaptic Plasticity in the Dorsolateral Striatum"

_ijms, 2024, doi:10.3390/ijms25010581_

Round 1
Reviewer 1 Report
Comments and Suggestions for Authors
The aim of the study is to evaluate how astrocytes can influence eCB-LTD in striatum and, to reach this aim, authors used different approaches to inhibit astrocyte function. They demonstrated that astrocytes are able to modulate synaptic transmission and to influence synaptic plasticity in the form of eCB-LTD. In particular authors showed that when astrocyte function was inhibited with FC or CNO a synaptic depression of PS and a facilitation of HFS induced LTD were evident. In addition, FC induced reduction of PS was nor inhibited by AM251, a cb1R antagonist, as observed in a previous paper, suggesting no involvement of CB1 receptor in FC or CNO induced reduction of PS. These results are clear but the conclusions of authors don’t appear convincing. Its known that in striatum, HFS induced LTD is eCB dependent, the authors advances the hypothesis that the facilitatory role of FC on LTD is linked to a direct modulation of eCB signalling (line 283-284….affecting the production, release or degradation of eCBs) but no data supporting this hypothesis emerges in the paper. HFS-LTD could be modulated by other neurotransmitters.
Probably glutamate has a pivotal role in astrocytes induced HFS-LTD modulation as suggested in the discussion (in a previous paper authors demonstrated an increase in glutamate level following astrocytes function block), but dopamine could also be involved: dopamine levels increase after FC application and FC induced synaptic depression is blocked in presence of sulpiride (previously demonstrated by Adermark et al 2021), in addition dopamine is critical for eCB release and for triggering eCB-LTD as demonstrated by Kreitzer and Malenka 2007.
Major revision:
1) Based on the aim of the paper, the introduction should be integrated with information and references on the role of astrocytes in synaptic plasticity
2) In the paper the author assumes that FC induced facilitation of HFS-LTD is due to a modulation of eCB signalling, they should provide direct evidence of astrocyte’s involvement in eCB metabolism or release.
3) Authors should discuss their result taking in consideration data from their previous paper (Adermark et al 2021) and data by Cavaccini et al 2020, on the role of astrocytes in synaptic transmission in striatum.
4) Could dopamine be responsible of FC induced HFS-LTD modulation through its ability to modulate eCB levels?
5) In fig 1C, FC trend, after HFS stimulation is constantly rising, please prolong the observation to 1 hour to show no recovery to the level of aCSF when HFS is applied in presence of FC.
Minor revision:
Please enter the timing of application of a drug and when a drug was applied respect to another one (an example: when AM251 was applied respect to FC? and Win? How long after FC was Win applied?)

Author Response
We appreciate the comments raised by the reviewers and have now performed a substantial revision of the manuscript. Apart from increased introduction and discussion sections (substantial increase in words and references), we have also performed additional experiments and immunohistochemistry. All changes in the manuscript are marked in blue, and responses to each issue is outlined below. We hope you will consider this revised manuscript acceptable for inclusion in the special issue “Mechanisms of the
Endocannabinoid System and Their Role in Neurophysiology and Psychopathology”.
Major points:
1) Fluorocitrate (FC) experiments: It is known that this metabolite is preferentially transported by astrocytes, however their effects on the metabolism of astrocytes are not fully understood. The effects of FC are dose dependent, and it has been shown to alter the glutamate – glutamine metabolism, glycogen content and mitochondrial membrane potential but without altering the ATP levels (Ref 36 of manuscript and others). To my knowledge, there are almost no studies that carefully studied the effects of FC on glycolytic or mitochondrial fluxes, or the effects on critical astrocyte-derived metabolites as lactate or L-serine (precursor of D-serine, co-agonist of synaptic NMDARs). Thus, the notion that FC treatment results in “tissue devoid of metabolically active astrocyte” (line 286) is an overstatement that needs to be considered for the data presentation and interpretation. For example, Fig 1G depicts the inhibition of glutamate uptake, but how can this be explained if ATP levels remain stable? Similarly, the panel depicts inhibition of mitochondria, but the current literature does not fully support the inhibition of ALL mitochondria functions.
Regarding the data interpretation, authors claim that FC facilitates HFS-LTD by promoting eCB signaling, however the data could also be interpreted as a failure of the astrocyte-neuron metabolic coupling, in other words, FC might alter the ability of astrocytes to supply metabolites necessary for neuronal function as glutamine, L-serine or lactate, which in turn result in altered neuronal function. This might be plausible as the HFS in presence of FC result in filed potentials close to zero. Regarding the effect of FC in presence of WIN55, authors interpret this result as “ …astrocytic regulation of eCB signaling is upstream from presynaptic CB1R activation” (lines 167-168”. The data do not support this conclusion, as FC might also alter neurotransmission by a parallel pathway that does not interact with ECS signaling. Therefore, I am not fully convinced that FC treatment demonstrates a link between astrocyte energy metabolism and the ECS signaling.
We appreciate the comment raised by the reviewer and have now substantially increased the discussion regarding the influence displayed by FC on astrocyte function and the different targets this compound may have. We have also adjusted the figure to clarify that FC affects the TCA cycle in the mitochondria. We have also included that the effects are dose-dependent, and that while ATP levels have been reported to be reduced in many studies, only minor effects may be present when applying the concentration used here. While not monitored here, FC-mediated effects on ATP appears to depend on neuronal activity/stimulation, and since slices were repeatedly stimulated in the field potential recordings, this might however have enhanced the impact by FC on ATP levels.
We now clearly state that part of the result may be associated with the reduced supply of metabolites necessary for neurotransmission, and that we did not measure this here. However, we have included additional experiments demonstrating that HFS-LTD in FC treated slices is CB1R dependent. While FC still might alter neurotransmission by a parallel pathway, this pathway still involves eCB signaling.
2) DREADDs strategy: Authors need to be careful with the use of “excitatory” (Gq) and “inhibitory” (Gi) DREADDs. While the downstream signaling cascades triggered by these DREADDs are different, both can modulate astrocytes calcium dynamics (REF). By considering that astrocytes calcium dynamics are a prime regulator of astrocyte function, it is not possible to consider a Gi-coupled DREADD as inhibitory in astrocytes. Thus, this point needs to be considered in the data and discussion. Also, the rationale of using different concentration of CNO needs to be described and explained. Finally, authors need to show data regarding the specificity and area of infection achieved by their viral approach to express the DREADDs.
We have now substantially increased the discussion regarding the use of Gq and Gi-coupled DREADDs and the limits with using this technique. We have also removed all phrases involving words as “excitatory” or “inhibitory” and only state Gq or Gi DREADDs. Since CNO depressed PS amplitude also in brain slices from sham-treated rats, we needed to lower the dose to avoid unspecific effects produced by CNO/DMSO on neurotransmission.
3) Carbenoxolone (CBX) experiments: Authors need to be careful with this pharmacological tool. Connexins are known to play a role in the control of astrocyte communication via calcium dynamics or control of K+ buffering, but also Connexins control the diffusion of metabolites as glucose and lactate (PMID 19056987 and 32332733) which might alter the astrocyte metabolic function and in turn neurotransmission. Furthermore, CBX is known to be unspecific (for example, PMID 33269468 and 15028741). I would recommend authors to test the gap junction inhibitor GAP 27 to confirm their data. Finally, authors need to consider the potential effects of gap junction blockade on the neurometabolic coupling of astrocyte and neurons.
We have now included the references suggested by the reviewer and have further increased the discussion regarding the lack of specificity displayed by gap junction blockers. Unfortunately, it is not clear for references discussing unspecific effects what concentrations of DMSO that could have been applied, but we have clarified that we used water soluble carbenoxolone disodium, which at least may reduce the risk of unspecific effects associated with DMSO. We also agree that additional experiments are recommended to further outline the role of gap junctions and now state this in the discussion. However, due to the time limit for returning a full revision, and the number of additional experiments suggested, we were not able to include these experiments here.
- APV experiments: Could the authors provide an explanation for the (almost) lack of effect of APV on the basal field potentials (Fig 4D)? Also, the effect of FC seems to be amplified by APV (~20% decrease on Fig 1 vs ~40% with APV+FC), however APV seems to not alter the field potentials. What is the explanation for this result?
While we stated in the result section that there was no significant difference between synaptic depression induced by APV+FC vs FC alone, we appreciate this comment since it highlights the risk of misinterpreting the heavily compressed graph (compared to figure 1) combined with the slight depression induced by APV itself. We have now performed additional experiments to further explore the putative potentiating effect displayed by APV, and also outlined the influence displayed by mGluR2/3s, which are considered presynaptic autoreceptors but also may be expressed by astrocytes. Neither APV not mGluR2/3R antagonist prevented (or facilitated) synaptic depression. To reduce the risk of misinterpreting the graph, we have now updated the figure.
Regarding the lack of effect on evoked PS amplitudes, this is not that surprising since the response reflects AMPA receptor activation. Evoked PS amplitudes in striatum are much more rapid as compared to hippocampus (around 2 ms), and this may contribute to the lower influence displayed by the NMDA component. In addition, aCSF contain Mg2+, MSNs are depolarized, and receptors are putatively further inhibited by glial released molecules such as kynurenic acid.
Minor points:
- Line 48. Several references regarding CB1 expression in astrocyte are missing, for example PMID 29480581 and 22385967. Please amend.
We have now included the recommended references.
- Line 49 – 49 The phrase “The crosstalk between neurons and astrocytes in eCB-mediated plasticity appears however to depend on the brain region studied”, could benefit from giving some examples with their corresponding references.
We have now increased this part of the introduction.
- Please indicate in the figure legends the number of slices and animals used.
We have now included this information where it was missing.
- Several Figure panels are not indicated in the text of results. Fig 1C or Fig 2A are not mentioned. Please amend accordingly.
We have amended accordingly.
5) Figure 2E. Is the statistics comparison correct? For what it is read on the text the difference is between sham vs “inhibitory DREADDs”.
The statistics in the graphs were related to the ANOVA performed on the three groups together. We have now altered according to the suggestion and only included the post-hoc tests.
- Carbenoxolone typos on line 223 and Fig 3A-B. Please amend.
Typos have been corrected.
- Line 248 – Typo “gliotransmittors”, please change to “gliotransmitters”.
Typos have been corrected.
8) Line 248. Please add a reference indicating that kynurenic acid is indeed a gliotransmitter (i.e., released by astrocytes upon neuronal activity/intracellular Ca2+ changes.
While kynurenic acid is released from astrocytes, and AMPA receptor stimulation enhances the release, the association between intracellular Ca2+ changes and subsequent release of kynurenic acid has to our knowledge not been clearly demonstrated. We have now rephrased to the “glia-derived molecule”.

Reviewer 2 Report
Comments and Suggestions for Authors
Please find attached the report.

Author Response

(The authors gave the same response as above.)

Reviewer 3 Report
Comments and Suggestions for Authors
Findings reported in this study conjure up a primary role for striatal astrocytes in regulating glutamatergic neurotransmission and plasticity. Impairment of astrocytes results in a long-lasting imbalance between excitation and inhibition.
Major points
Results obtained with the use of gap junction inhibitor carbenoxolone (CBX) may be called into question due to several shortcomings of CBX use. It is hardly soluble in water. The applied amount of DMSO as a solvent, however, may cause non-specific side effects. Than again, the more water-soluble CBX hemisuccinate may also act on succinate receptors. Authors are invited to repeat these experiments by applying connexin43 antibody to specifically inhibit astrocytic function.
|
Authors may want to explain findings by taking into account the astrocytic glutamate uptake-mediated GABA release from astrocytes. Minor points Non-explained abbreviations English style, typos, grammar |
Author Response
Results obtained with the use of gap junction inhibitor carbenoxolone (CBX) may be called into question due to several shortcomings of CBX use. It is hardly soluble in water. The applied amount of DMSO as a solvent, however, may cause non-specific side effects. Than again, the more water-soluble CBX hemisuccinate may also act on succinate receptors. Authors are invited to repeat these experiments by applying connexin43 antibody to specifically inhibit astrocytic function.
As stated in the method section (drugs) we used carbenoxolone disodium, which is dissoluble in H2O. Since this was not fully clear, we now write out “carbenoxolone disodium” throughout the manuscript.
https://www.tocris.com/products/carbenoxolone-disodium_3096
We have also increased the discussion regarding the specificity of gap junction blockers and state that additional experiments using other more selective gap-junction blockers are required to confirm these findings. However, due to the short time-frame for resubmitting the revision (ten days), and the number of additional experiments required, we were not able to perform the suggested experiments. We do however hope that the major revision performed with regards to text editing is satisfactory.
Authors may want to explain findings by taking into account the astrocytic glutamate uptake-mediated GABA release from astrocytes.
We have now included a statement concerning the putative involvement of GABAergic neurotransmission in contributing to these results. We were however not able to outline the putative involvement of this system but instead focused on glutamatergic signaling.
Minor points
Non-explained abbreviations
We hope all abbreviations have been explained.
English style, typos, grammar
We hope all typos are corrected.
Round 2
Reviewer 1 Report
Comments and Suggestions for Authors
I recommend to the author to review point 2 in the annex

Author Response
Reviewer comment no 2: In the paper the author assumes that FC induced facilitation of HFS-LTD is due to a modulation of eCB signalling, they should provide direct evidence of astrocyte’s involvement in eCB metabolism or release.
Response 1: We agree with the reviewer that this is a putative explanation and have now increased this discussion further in the manuscript. We have also performed additional experiments demonstrating that HFS-LTD during FC administration is blocked by CB1R antagonist, suggesting that synaptic depression induced by gliotransmitter release (other than eCBs) most likely not contributes to the facilitation of HFS-LTD.
Response Second revision: We might have misunderstood the issue raised by the reviewer, but now demonstrate that the potentiation of HFS-LTD is eCB-mediated, thus not associated with concomitant LTD induced by for instance adenosine. We believe that the enhanced eCB signaling is connected to increased calcium influx in neurons, thereby facilitating the production of eCBs. However, others have demonstrated that astrocytes degrade eCBs and if astrocytes are impaired, enhanced signalling may be connected to reduced metabolism of eCBs and thus elevated circulating levels. Lastly, astrocytes affect neurotransmission, and considering that eCB signaling involve a postsynaptic release step that is dependent on synaptic activity this could also affect the amount of eCBs released. However, since the potentiation was prevented by NMDA receptor antagonist, and NMDAR are highly permeable to calcium, we suggest that this finding supports the idea that calcium influx in neurons is enhanced thereby stimulating eCB formation. We have also included a reference demonstrating that eCB-mediated LTD in striatum can be blocked by chelation of calcium in the patch clamped MSNs, thereby demonstrating that eCBs are primarily released from the postsynaptic neuron. Since the tools available is insufficient to fully prove this mechanism of action, we also discuss other signalling pathways that may be recruited to promote HFS-LTD on lines 502-528. In this second revision, we have also specified that while FC-mediated synaptic depression in the striatum is independent on GABAA receptor activation, GABAergic neurotransmission may still play a role in facilitation of HFS-LTD.
In addition, since the reviewer did not approve with the conclusion, we have now removed the speculative sentence from the abstract, where we suggested that postsynaptic calcium could be enhanced thereby resulting in enhanced eCB signaling, and instead further emphasise the limitations of our study.
Throughout the manuscript we point to the fact that it is hard to disentangle neurons from astrocytes, and that no tool can be considered fully specific. Still, all our approaches generated similar results and thus collectively support a role of astrocytes in modulating neurotransmission and plasticity. If the adjustments made in the manuscript are not what the reviewer had in mind, we hope that the issue can be more specified so that we can make the appropriate revisions.
Reviewer 2 Report
Comments and Suggestions for Authors
The authors have replied to all the concerns raised, either by introducing new data or improving data interpretation and discussion.
I just have a minor comment:
On line 256 “3.2 Chemogenetic inhibition…” Authors forgot to change this sub-title to avoid the use of terms related to “inhibitory DREADDs activity”. Changing the title to “Chemogenetic manipulation of astrocyte function…”or something similar, should solve this point.
Author Response
We are thankful that the reviewer noticed this mistake, and has now changed accordingly.
(3.2. Activation of Gi-coupled DREADDs targeting astrocytes...)
Reviewer 3 Report
Comments and Suggestions for Authors
Correct reference to "Kilb, W., Kirischuk, S., 2022. GABA Release from Astrocytes in Health and Disease. Int J Mol Sci 23." in the the MS text: Kilb and Kirischuk".
It is to note, that this reference is not the primary source of information for the involvement of Glu uptake induced astrocytic GABA release.
Author Response
We have now removed the review from 2022 and instead included an original study from Hertz and Schousboe from 1978, and the study from Heja et al from 2009 more focusing on the association between glutamate uptake and GABA release.
Round 3
Reviewer 1 Report
Comments and Suggestions for Authors
the issue n°2 is considered solved